# Neuron Shapley: Discovering the Responsible Neurons

**Amirata Ghorbani**
Department of Electrical Engineering
Stanford University
Stanford, CA 94025
amiratag@stanford.edu

**James Zou**[*]
Department of Biomedical Data Science
Stanford University
Stanford, CA 94025
jamesz@stanford.edu

## Abstract

We develop Neuron Shapley as a new framework to quantify the contribution of individual neurons to the prediction and performance of a deep network. By accounting for interactions across neurons, Neuron Shapley is more effective in identifying important filters compared to common approaches based on activation patterns. Interestingly, removing just 30 filters with the highest Shapley scores effectively destroys the prediction accuracy of Inception-v3 on ImageNet. Visualization of these few critical filters provides insights into how the network functions. Neuron Shapley is a flexible framework and can be applied to identify responsible neurons in many tasks. We illustrate additional applications of identifying filters that are responsible for biased prediction in facial recognition and filters that are vulnerable to adversarial attacks. Removing these filters is a quick way to repair models. Computing exact Shapley values is computationally infeasible and therefore sampling-based approximations are used in practice. We introduce a new multi-armed bandit algorithm that is able to efficiently detect neurons with the largest Shapley value orders of magnitude faster than existing Shapley value approximation methods.

## 1 Introduction

Understanding and interpreting the behavior of a trained neural net has gained increasing attention in the machine learning (ML) community. A popular approach is to interpret and visualize the behavior of specific neurons (People often randomly selected neurons from different layers to visualize.[35]) in a trained network , and there are several ways of doing this. For example, showing which training data leads to the most positive or negative activation of this neuron, or performing a deep-dream style visualization to modify the input to greedily activate the neuron [35]. Such approaches are widely used and can give interesting insights into the function of the network. However, because networks typically have a large number of neurons, analysis of individual neurons is often ad-hoc and it's typically not clear which ones to investigate. Moreover, a neruon's relevance to the overall function of network or its interactions with all other neurons is not considered.

In this paper, we propose a new framework to address these limitations by systematically identifying the neurons that are the most *important* contributors to the network's function. The basis of this framework is a new algorithm we propose—Neuron Shapley—for quantifying the importance of each neuron in a trained network while accounting for complex interactions between neurons. In addition, the algorithm utilizes multi armed bandit to perform this task efficiently. Interestingly, for several standard image recognition tasks, a small number of fewer than 30 neurons (filters) are crucially

---

[*]Corresponding author.

necessary for the model to achieve good prediction accuracy. Interpretation of these critical neurons provides more systematic insights into how the network functions.

Most importantly, Neuron Shapley is a very flexible framework and can be applied to a wide group of tasks beyond model interpretation. When applied to a facial recognition model that makes biased predictions against black women, Neuron Shapley identifies the (few) neurons that are responsible for this disparity. It also identifies neurons that are the most responsible for vulnerabilities to adversarial attacks. This opens up an interesting opportunity to use Neuron Shapley for fast model repair without needing to retrain. For example, our experiments show that simply zeroing out the few "culprit" neurons reduces disparity without much degradation to the overall accuracy.

**Our contributions** We summarize the contributions of this work here. **Conceptual**: we develop the Neuron Shapley framework to quantify the contribution of each neuron to an arbitrary performance aspect of a network. **Algorithmic**: we introduce a new multi-arm bandit based algorithm that efficiently estimates Neuron Shapley values in large networks. **Empirical**: our systematic experiments discover several interesting findings, including the phenomenon that a small number of neurons are critical to different aspects of a network's performance, e.g. accuracy, fairness, robustness. This facilitates both model interpretation and repair.

**Related literature** Shapley value [37] has been studied in the cooperative game theory and economics [38] as a fair distribution method. In recent years, Shapley value has been widely adopted in machine learning interpretability research. Interpretability literature and its use of Shapley value can be divided in three groups: 1- *Feature-importance* methods that compute the contribution of each input feature to network's performance where Shapley value is naturally desirable as it considers the interactions among input features. Research has focused on efficient approximation methods and developing model-specific methods. [21, 27, 26, 31, 7]. 2- *Sample-importance* methods that compute the contribution of each training example [18, 13] where Shapley value is used to find the fair value of each datum. 3- *Element-importance* methods to compute the contribution of individual elements of a machine learning model e.g. neurons in a deep network. Although computing Shapley value of neurons has been explored for pruning relatively small models [41], to the best of our knowledge, Neuron Shapley is the first algorithm that can efficiently compute Shapley values for arbitrarily large models across various interpreatbility and model repair tasks. A trivial approach for finding the importance of neurons is to act as if the neurons in a given layer are the input features and apply a feature-importance method to neurons in that layer. Unfortunately, the resulting importance score is not desirable as it does not incorporate the interaction of neurons that are not in the same layer.

Gradient-based interpretability methods have been also been developed to address feature, sample and element importance [34, 4, 1, 39]. For computing the importance of neurons, a group of methods have been proposed that are mostly based on extending feature-importance algorithms like Integrated Gradients [42] to neurons [9, 23]. While very useful, these methods have several drawbacks: the scores are limited to interactions of neurons in nearby layers and do not satisfy the fair credit allocation properties; they are mostly curated for computing neuron contributions to a specific sample's prediction and not the global behavior of the model; and unlike Neuron Shapley that can be applied to any task, they are limited to cases where one can take gradients of the task with respect to neurons' activations.

Post-hoc model repair has recently been studied in the specific case of fairness applications, but these typically involve some retraining of the network [20]. Repair through Neuron Shapley has the benefit of not requiring retraining, which is faster and is especially useful when access to a large amount of training data is hard for the end-user of the network. Finally, we introduce an adaptive multi-arm bandit algorithm for efficient estimation of the Shapley value. To the best of our knowledge, this is the first work to utilize multi-armed bandit for efficient approximation of Shapley value. This algorithm enables us, for the first time, to compute Neuron Shapley values in large-scale state-of-the-art convolutional networks.

## 2 Shapley Value for Neurons

**Preliminaries:** We work with a trained ML model $\mathscr{M}$ which is composed of a set of $n$ individual elements: $N = \{m_i\}_{i=1}^n$ e.g. $\mathscr{M}$ could be a fully convolutional neural network with $L$ layers each with $n_{l \in \{1,...,L\}}$ filters. The elements are its $n$ filters where $n = \sum_{l=1}^L n_l$. We focus on convolutional networks in this paper as most of the interpretation methods are for image data. The Neuron Shapley

approach is also applicable to other network architectures as well as to other ML models such as Random Forests. The trained model is evaluated on a specific performance metric $V$. In our setting $V$ is a black-box that takes any network as input and returns a score e.g. accuracy, loss, disparity on different minority groups, etc. The performance on the full model is denoted as $V(N)$. Our goal is to *assign responsibility to each neuron* $i \in N$. Mathematically, we want to partition overall performance metric $V(N)$ among model elements: $\phi_i(V, N) \in \mathbb{R}$ is $m_i$'s contribution towards $V(N)$ such that $\sum_{i=1}^{N} \phi_i(V, N) = V(N)$. For simplicity, we use $\phi_i$ to denote $\phi_i(V, N)$.

In order to evaluate the contribution of elements (neurons) in the model, we would like to "zero out" these elements. In a convnet, this is done by fixing the output of a filter to be its mean output for set of validation images. This will kill the flow of information through that filter while keeping the mean statistics of the propagated signal from that layer intact (which is not the case if the output is replaced by all zeros). We are interested in subsets $S \subseteq N$ (i.e. sub networks), and we write $V(S)$ to denote the performance of the model after all elements in $N \setminus S$ are zeroed out. Note that we do not retrain the model after certain elements are zeroed out (all of the weights of the network are fixed). We simply take the modified network and evaluate its test performance $V(S)$. The reason for doing this is that even fine-tuning the network for each $S$ would be prohibitively expensive computationally.

**Desirable properties for neuron valuation** For a given model and performance metric, there are many ways to partition $V(N)$ among the elements in $N$. This task is further complicated by the fact that different neurons in a network can have complex interactions. We take an axiomatic approach and use the Shapley value to compute $\phi_i$ because Shapley value is the unique partition of $V(N)$ that satisfies several desirable properties. We list these properties below:

- **Zero contribution** One decision to make is how to handle neurons that have no contribution. We say that a neuron $i$ has no contribution if $\forall S \subseteq N \setminus \{i\} : V(S \cup \{i\}) = V(S)$. In words, this means that it does not change the performance when added to any subset of other neurons in the network. For such null neurons, the valuation should be $\phi_i = 0$. One simple example is a neuron with all-zero parameters.

- **Symmetric elements** Two neurons should have equal contributions assigned if they are exchangeable under every possible setting: $\phi_i = \phi_j$ if $\forall S \subseteq N \setminus \{i, j\} : V(S \cup i) = V(S \cup j)$. Intuitively if adding $i$ or $j$ to any subnetwork gives the same performance, then they should have the same value.

- **Additivity in Performance Metric** In many practical settings, there are two or more performance metrics $V_1, V_2, \ldots$ for a given model. For example $V_1$ measures it's accuracy on one test point and $V_2$ is its accuracy on a second test point. A natural way to measure the overall performance of the model is having a linear combination of such metrics e.g. $V = V_1 + V_2$. We would like each neuron's overall contributions to follow this linear relationship i.e. $\phi_i(V, N) = \phi_i(V_1, N) + \phi_i(V_2, N)$. A real-world example of additivity's importance is a setting where computing the overall performance is not an option due to privacy concerns e.g. a healthcare ML model deployed at several hospitals. The hospitals are not allowed to share their test data with us and can only compute and report each neuron's contribution to their own task. Additivity allows us to gather the local contributions and aggregate them.

The contribution formula that uniquely satisfies all these properties is: $\phi_i = \frac{1}{|N|} \sum_{S \subseteq N - \{i\}} (V(S \cup \{i\}) - V(S)) / \binom{|N|-1}{|S|}$ , i.e. a neuron's contribution is its marginal contribution to the performance of every subnetwork $S$ of the original model (normalized by the number of subnetworks with the same cardinality $|S|$). Most importantly, this formula takes into account the interactions between neurons. As a simple example, suppose there are two neurons that improve performance only if they are both present or absent and harm performance if only one is present. The equation considers all these possible settings. This, to our knowledge, is one of the few methods that take such interactions into account and is inspired by similar approaches in Game Theory [15].

This formula is equivalent to the Shapley value [37, 38] originally defined for cooperative games. In a cooperative game, $N$ players are related to each other through a score function $v : 2^n \to \mathbb{R}$ where $v(S)$ is the reward if players in $N \setminus S$ opt out. Shapley value was introduced as an equitable way of sharing the group reward among the players where equitable means satisfying the aforementioned properties. In our context, we will refer to $\phi_i$ as **Neuron Shapley**. It's possible to make a direct

mapping between our setting and a cooperative game; therefore, proving the uniqueness of Neuron Shapley. The proof is discussed in Appendix B.

## 3   Estimating Neuron Shapley

While Neuron Shapley has desirable properties, it is computationally expensive to compute. Since there are exponential number of sets $S$, Computing the Shapley value $\phi_i$ requires an exponential number of operations. In what follows, we discuss several techniques for efficiently approximating Neuron Shapley. We first rephrase the computational problem of Shapley values to a statistical one. We are then able to introduce approximation methods that result in orders of magnitude speed-ups.

**Monte-Carlo estimation**   For a model with $N$ elements, the Shapley value of the $i$'th component could be written as [13]: $\phi_i = \mathbb{E}_{\pi \sim \Pi}[V(S_\pi^i \cup \{i\}) - V(S_\pi^i)]$ , where $\Pi$ is a uniform distribution over $N!$ permutations of the model elements and $S_\pi^i$ is the set of elements that appear before the $i$'th one in a given permutation $\pi$ (empty set if $i$ is the first).

In other words, approximating $\phi_i$ is equivalent to estimating the mean of a random variable. Therefore, for an chosen error bound, Monte-Carlo estimation can give an unbiased approximation of $\phi_i$. Error analysis of this method of approximation has been studied in the literature [30, 6, 29].

**Early Truncation**   The main computational expense in the equation above is computing the marginal contribution of every element: $V(S_\pi^i \cup \{i\}) - V(S_\pi^i)$. For early elements in $\pi$, $S_\pi^i$ is small. For a small enough $S_\pi^i$, the model's performance $V(S_\pi^i \subseteq N)$ degrades to zero or negligible as a large number of connections in the network are removed (see Appendix D for examples). By utilizing this fact, for a sampled permutation $\pi$, we can abstain from computing the marginal contribution of elements that appear early on i.e. $\phi(S_\pi^i \cup \{i\}) - \phi(S_\pi^i) \approx 0$ if $S_\pi^i$ is small. In this work, we define a "performance threshold" below which the model is considered dead. This truncation can lead to substantial computational savings (close to one order of magnitude).

**Adaptive Sampling**   In most of our applications, we are more interested in accurately identifying the important contributing neurons in the model rather than measuring their exact Shapley values. This is particularly relevant since, as we will see, there is typically only a sparse number of influential neurons while the values for the rest are close to zero. Algorithmically, our problem is simplified to finding the subset[2] of bounded random variables with the largest expected value from a set of $n$ bounded random variables. This can be formulated as a multi-armed-bandit (MAB) problem which has been successfully used in other settings to speed up computation [2, 17, 24, 46].

The MAB component of the algorithm is described in Alg. 1 and we explain the intuition here. For each neuron in the model, we keep tracking a lower and upper confidence bound (CB) on its value $\phi_i$, which comes from standard estimation bounds. The goal is to confidently detect the top-$k$ neurons. Therefore, at each iteration, instead of sampling the marginal contribution of all neurons (as is the case for standard Monte Carlo approximation the above equation), we only sample for the subset of neurons where the $k$'th largest value at that iteration is in between their lower and upper bounds. If there are no neurons satisfying the sampling condition, it means that the top-$k$ neurons are confidently separated (up to an error tolerance) from the rest. We show in Appendix G that adaptive sampling results in a nearly one order of magnitude speedup. Combining the three approximation methods, we introduce a novel algorithm that we refer to as "Truncated Multi Armed Bandit Shapley" (TMAB-Shapley) described in Alg. 1.

## 4   Experiments & Applications

**Implementation details**[3]**:**  We apply Neuron Shapley to two famous convolutional neural network architectures. First is the Inception-v3 [44] architecture trained on the ILSVRC2012 (a.k.a ImageNet) [36] dataset (reported test accuracy 78.1%). We use Alg. 1 to compute the Neuron Shapley value for each of the 17216 filters preceding the logit layer in this network. We divide the released ImageNet validation set into two parts (25000 images each) to serve as validation and test sets. The second model is the SqueezeNet [16] architecture with 2976 filters that we trained on the celebA [25] dataset to detect gender from face images (98.0% test accuracy). In all the experiments, we set

**Algorithm 1 Truncated Multi Armed Bandit Shapley**

1: **Input:** Network's elements $N = \{1, \ldots, n\}$; performance metric $V(.)$; failure probability $\delta$, tolerance $\epsilon$, number of important elements $k$, Early truncation performance $v_T$
2: **Output:** Shapley value of elements: $\{\phi_i\}_{i=1}^n$
3: **Initializations:** $\{\phi_i\}_{i=1}^n = 0, \{\sigma_i\}_{i=1}^n = 0, \mathscr{U} = N, t = 0$
4: **while** $\mathscr{U} \neq \emptyset$ **do**
5:     $t \leftarrow t + 1$
6:     Random permutation of network's elements: $\pi^t = \{\pi^t[1], \ldots, \pi^t[n]\}$
7:     $v_0^t \leftarrow V(N)$
8:     **for** $j \in \{1, \ldots, N\}$ **do**
9:       **if** $j \in \mathscr{U}$ **then**
10:         **if** $v_{j-1}^t < v_T$ **then**
11:           $v_j^t \leftarrow v_{j-1}^t$
12:         **else**
13:           $v_j^t \leftarrow v(\{\pi^t[j+1], \ldots, \pi^t[n]\})$
14:         $\phi_{\pi^t[j]}, \sigma_{\pi^t[j]} \leftarrow \text{Moving Average}(v_{j-1}^t - v_j^t, \phi_{\pi^t[j]}), \text{Moving Variance}(v_{j-1}^t - v_j^t, \phi_{\pi^t[j]})$
15:         $\phi_{\pi^t[j]}^{ub}, \phi_{\pi^t[j]}^{lb} \leftarrow \text{Confidence Bounds}(\phi_{\pi^t[j]}, \sigma_{\pi^t[j]}, t)$
16:     $\mathscr{U} \leftarrow \{i : \phi_i^{lb} + \epsilon < k\text{'th largest } \{\phi_i\}_{i=1}^n < \phi_i^{ub} - \epsilon\}$

$k = 100$ to detect the top-100 important filters. The results are robust to the choice of $k$. We use empirical Bernstein [33, 32] to compute the confidence bounds in Alg. 1.

**TMAB-Shapley is efficient** We run the original MC-Shapley algorithm and our TMAB-Shapley algorithm on the Squeezenet model ($\delta = 0.05, \epsilon = 10^{-4}$). It turns out that on average, TMAB-Shapley requires around one order of magnitude fewer samples. Fig. 3(a) shows the histogram of number of samples each algorithm takes to compute the value of filters. It can be observed that while MC-Shapley requires on the order of $100k$ samples, TMAB-Shapley converges with less than $10k$ samples for most of the filters. Empirically, although TMAB-Shapley is not optimized for accurate estimation of Shapley values, the resulting values are very close to the more accurate output of MC-Shapley ($R^2$=0.975, Rank Corr.= 0.988). In addition, the truncation trick results in another order of magnitude speed-up (Appendix D).

**Neuron Shapley identifies a small number of critical neurons** We apply Alg. 1 to compute the Neuron Shapley value for all of the Inception-v3 filters. Here, we used the performance metric of the overall accuracy of the network (on a randomly sampled batch of images) as $V(.)$. Interestingly Neuron Shapley values are very sparse. We can evaluate the impact of the neurons with the largest Shapley values by zeroing them out. Removing just the top 10 filters and the overall test accuracy of Inception-v3 dropped from 74% to 38%; removing the top 20 neurons and the accuracy dropped to 8%. It is interesting that a handful of neurons have such a strong effect on the network's performance. In contrast, removing 20 random neurons in the network does not change the accuracy of Inception-v3. A related phenomenon has shown that many connections/weights in the network can be removed without damaging performance [10]. The difference is that the pruning literature has primarily focused on removing connections, while our experiment here is for removing neurons.

The sparse set of critical filters as identified by high Shapley values is a natural set of neurons to visualize and interpret. In Fig. 1, we visualize the filter with the highest Shapley value in 7 of the layers of Inception-v3. We provide two types of visualizations: 1) Deep Dream images (first column of each block)[4]; 2) and the five images in the validation set that result in the most positive or most negative activation of the filter. The critical neurons in the earlier layers capture color (white vs. black) and texture (vertical stripes vs. smooth). The later layer critical neurons capture more complex concepts like colorfulness or crowdedness of the image which is consistent with previous findings using different approaches [19, 3, 12]. The final component of Fig. 1 shows how the top 100 filters with the highest Shapley values are distributed in different layers of Inception-v3. Overall more of these filters tend to be in the early layers, consistent with the notion that initial layers learn general

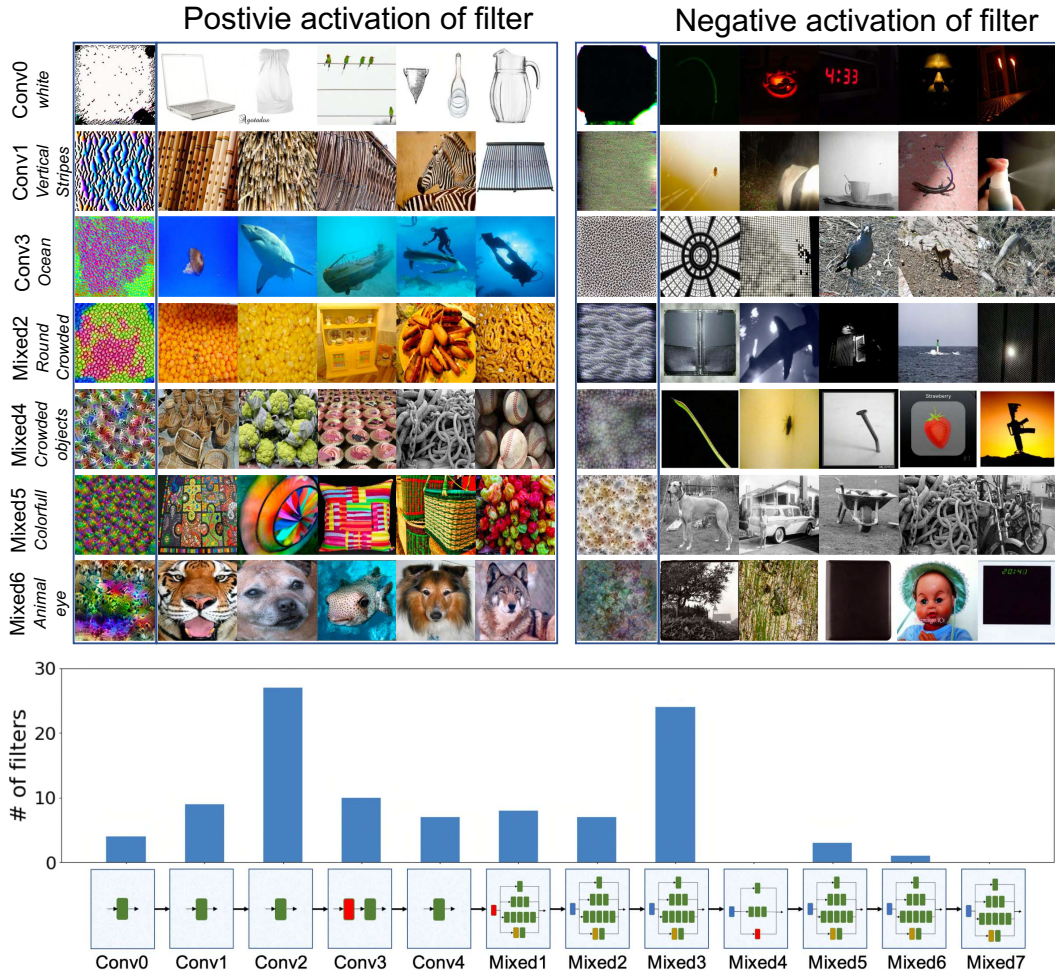

Figure 1: **Visualizing filters critical for overall accuracy** We visualize the highest Shapley value filters for a select few Inception-v3 blocks. For each filter, we show 5 example images that activate that neuron most positively or most negatively. Additionally, we optimize a random input to highly activate (positively or negatively) the selected neuron. These filters can have meaningful interpretations, which we write on the left. Earlier layer filters extract simple features like color or pattern. As we go deeper, filters capture sophisticated features like crowdedness or how much color is in the image. On the bottom, we show how many of the top-100 contributing filters appear in each layer.

concepts and the deeper layers are more class-specific. More results are discussed in Appendix E. We report similar experiment results for the Squeezenet model in Appendix F.

**Class specific critical neurons** We can dive more deeply to investigate which neurons specifically work for prediction of a chosen class. For a given class (e.g. zebra, dumbbell), we use class recall as the performance metric $V$. We apply Alg. 1 and make sure the top-k neurons are class specific by excluding the top-20% neurons that contributed mostly to the overall accuracy from above experiment (no MC sampling). The result are the top-100 neurons that highly contribute to the chosen class and are not crucial for the overall performance of the model.

As before, Neuron Shapley discovers that for every class, there exists a small number of filters that are critical. We provide results for four representative classes (carousel, zebra, police van and dumbbell) in Fig. 2. In each of these classes removing top 40 filters leads to a dramatic decline in the network's ability to detect that class (Fig. 2(a)) while the overall performance of the model remains intact. For comparison, we also applied 4 popular alternative approaches for identifying important neurons—by

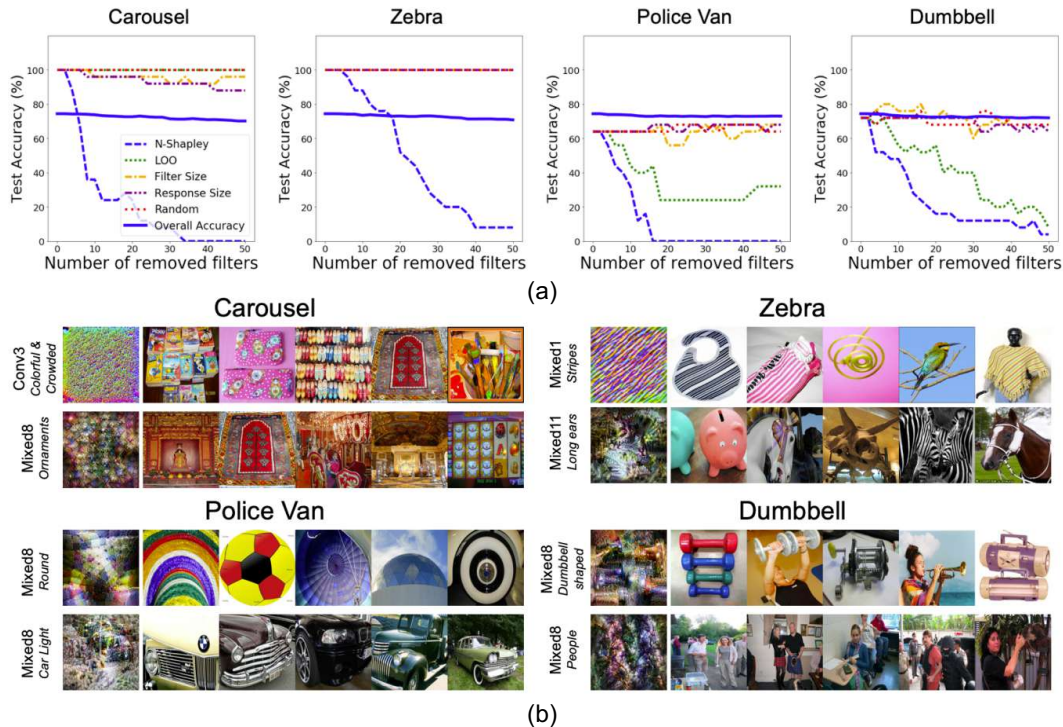

Figure 2: **Class-specific critical neurons** (a) Removing filters with the highest class-specific Shapley values (blue dash) reduce the class prediction accuracy more effectively than removing filters identified by other approaches. We select four representative classes to show. (b) We visualize two critical filters for each class by showing the top 5 most positively activating images along with the deep dream visualization of the filter. (c) Class-specific filters are more common in the deeper layers.

filter size ($\ell_2$ norm of the weights), $\ell_2$ norm of the filter's response, leave-on-out impact (drop in $V(.)$ if just that filter is zeroed out). Overall, Neuron Shapley is more effective in finding critical neurons. We further report the network's overall accuracy across all the classes; removing class-specific critical neurons does not affect the overall performance.

Fig. 2 visualizes two of the most critical neurons for each of the four classes—Deep Dream and the top five highest activating training images are shown for each filter. For the zebra class, diagonal stripes is a critical filter. For dumbbell, one critical neuron clearly captures dumbbell like shapes directly; the second captures people with arms visible, likely because that's highly correlated with dumbbells in natural images (which is observed in previous literature [19, 12]).

Neuron Shapley is a flexible framework that can be used to identify neurons that are responsible for many types of network behavior beyond the standard prediction accuracy. We illustrate its usage on two important applications in fairness and adversarial attacks.

**Discovering unfair filters** It has been shown that the gender detection models have certain biases towards minorities [5]; for example, they are less accurate on female faces and especially on black female faces. We took SqueezeNet trained on CelebA faces and use its accuracy on the PPB dataset [5] as the evaluation metric $V$. PPB dataset has an equal representation of four subgroups of gender-race [5] and a model's accuracy on PPB is a good measure of its fairness. Alg. 1 is used to compute the Shapley value of each filter in SqueezeNet. In this case, we are most interested in the filters with the *most negative* values as they would decrease fairness and contribute to the disparity. Zeroing out these "culprit" filters greatly increased the gender classification accuracy on black female (BF) faces from 54.7% to 81.9% (Fig. 3). It also led to a substantial improvement for white females (WF). The average accuracy on PPB increased from 84.9% to 91.7%. The performance on the original CelebA data only dropped a little from this modification. This suggests the interesting potential of using Neuron Shapley for rapid model repair. Zeroing out filters can be much faster and easier than retraining the network, and it also does not require an extensive training set.

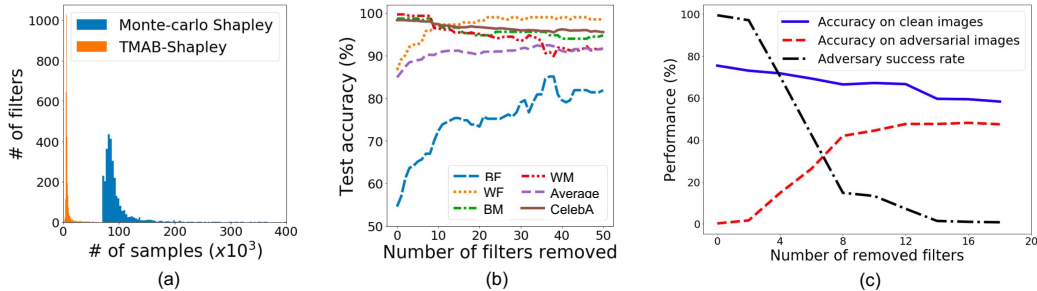

Figure 3: (a) The y-axis shows the number of filters that required the given number of samples on x-axis in each algorithms. Using Alg. 1, most filters require around 10 times fewer samples compared to MC-Shapley. (b) After computing each filter's contribution to the model's fair perfromance, we remove the ones with the most negative contribution. The model improves especially for black females (BF). The four populations are white female (WF), black female (BF), white male (WM) and black male (BM). (c) We compute each filter's contribution to the adversary's success rate. After removing most contributing neurons, the adversary is much less successful (black), and the model becomes able to detect a large portion of adversarial perturbed images as their true class (red).

**Identifying filters vulnerable to adversaries** An adversary can arbitrarily change the output of a deep network by adding imperceptible perturbations to the input image [14, 11]. We apply Neuron Shapley to identify filters that are most vulnerable to attacks in the Inception-v3 model. We use iterative PGD attack [22, 28] as an adversary whose goal is to perturb each validation image so it's misclassified as a randomly chosen class. Following literature [45], we use $\ell_\infty$ perturbations with size $\epsilon = 16/255$. For the performance metric we have "$V =$ Adversary's success rate $-$ Accuracy on clean images" where adversary's success rate adversary is the rate of fooling the network into predicting the randomly chosen labels. Therefore, a high Neuron Shapley value suggests that the filter is more targeted and leveraged by the adversary to produce misclassification. The rank correlation between these adversarial Shapley values and the original prediction accuracy Shapley values is just 0.3. This suggests that the network filters interact differently on the adversarially perturbed images than on the clean images.

After zeroing out the filters with the top 16 Shapley values (most vulnerable filters), the adversary's attack success rate drops from nearly 100% to nearly zero (0.1%). While the model's performance on clean images drops more moderately from $74\%$ to $67\%$. We recognize that this is not an effective defense method against white box attacks; while the modified network is robust to the *original* adversary, it is still vulnerable to a new white-box adversary specifically desigining to attack the modified network knowing which neurons were zeroed out. However, we investigated black-box adversaries—i.e. attacks from other architectures which are not used to compute the Neuron Shapley values. The modified network is also more robust against other black-box adversaries (i.e. attacks created by different architectures)—their attack success rate drops by $37\%$ on average. This suggests that Neuron Shapley can potentially offer a fast mechanism to repair models against black-box attacks without needing to retrain.

**Comparison to previous state-of-the-art** Here we compare Neuron Shapley with Neuron conductance, the recent state-of-the-art gradient-based method for estimating neuron importance [9]. Several works have extended Integrated Gradients [42] (a feature-importance method) to compute neuron importance [8, 9]. Neuron conductance was shown to have the best performance among all of these methods [9]. As we have seen for Inception-v3 on ImageNet, removing the top 30 filters with the highest Shapley value reduces the model's accuracy to random. It requires the removal twice as many filters based on conductance to achieve similar reduction. Next we compare the two methods head-to-head in the two model-repair experiments. For removing unfair filters, Neuron conductance increases test accuracy on PPB (fairness score) from $84.9\%$ to $88.7\%$ by removing 105 unfair filters. In comparison, Neuron Shapley increased the PPB performance to $91.7\%$ while removing many fewer filters. For discovering vulnerable filters to adversarial attacks, conductance needs to remove more filters (20 compared to 16 for Shapley) to achieve the same reduction in adversarial success rate. Moreover, removing the high conductance filters damaged the network much more compared to removing the high Shapley filters—the model's clean accuracy dropped to $53.2\%$ compared to

67% for Shapley. As a final comparison, we apply Neuron Conductance to one of the experiments in Fig 2. Removing top-10 and top-20 class specific neurons for the "Carousel" class reduced accuracy to $64\%$ and $40\%$ compared to $20\%$ and $8\%$ using Shapley (both methods use 25 images). All of the experiments demonstrate that Neuron Shapley more effectively identifies the sparse number of critical filters. This makes it better suited for interpretation and model repair. The two methods have similar computational expenses. We should also mention that both methods have nearly the same computational cost; that is, they equire on the same order of passes (forward or backward) through the network. More details are described in Appendix C.

## 5   Discussion

We introduce Neuron Shapley, a post-training method to quantify individual neuron's contribution to the network's performance. Neuron Shapley is theoretically principled due to its connection to game theory and is well-suited for disentangling the interactions of different neurons. We show that using Neuron Shapley, we are able to discover a sparse structure of critical neurons both on the class-level and the global-level. The model's behavior is largely dependant on the presence of the critical neurons. We can utilize this sparsity to apply post-training fixes to the model without any access to the training data; e.g. we can make the model more fair towards specific subgroups or less fragile against adversarial attacks just by removing a few responsible neurons. This opens interesting new approaches to model repair that deserves further investigation. A drawback of the Neuron Shapley formulation is its large computational cost. We have introduced a novel multi-arm bandit algorithm to reduce that cost by orders of magnitude. This enables us to efficiently compute Shapley values on widely used deep networks. Throughout this work, we have mainly focused on post-training edits to the model. One interesting future direction is to change the model given the neuron contributions and retrain it in an iterative fashion.

## Broader Impact

When a car breaks down, it is critical to know which component—the engine, tire, etc.—caused the issue. Similarly, when a machine learning model makes mistakes or does well, it is important to know which part of the model is responsible. In this work, we propose Neuron Shapley as a principled framework to quantify the contribution of every neuron to each prediction success and failure of the network. We propose an efficient adaptive algorithm for estimating this Shapley score. We further apply this approach to identify which neurons are responsible for predictions that are biased against minorities and neurons that are vulnerable to attacks. Neuron Shapley is thus a step toward making deep learning more accountable and responsible, which could have broad social impact. We suggest that it should be used in conjunction with other interpretation and analysis tools to provide a holistic assessment of the model.

## Footnotes

[2]The cardinality of this subset can be specified by the user or adaptively.

[3]Code is available on Github at `https://github.com/amiratag/neuronshapley`

[4]Deep Dream uses gradient ascent to directly optimize for activation of a filter's response while adding small transformations (jittering, blurring, etc) at each step [35].

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
