[Supplementary Material · NeuronShapley_appendix.pdf]



Figure 4: **Filter dropout effect** The Neuron Shapley results for two Squeezenet models trained on the celeb-A dataset, one trained with filter dropout and one without. (a) The histogram of values for the two models. (b) Removing the 30 highest value neurons shows that the dropout-trained model is more robust. (c) Removing filters with the least values shows that the dropout trained model is robust to the removal of almost half of the filters. The celebA test set has a 40-60 class imbalance; therefore we see the sharp drop from $60\%$ accuracy to $40\%$ accuracy.

## A  Dropout effect

The standard convnets, like the ones we use here, are typically trained without dropout regularization for conv layers [44, 43, 40]. We hypothesize that adding dropout could substantially change the Shapley values because it encourages filters to be more independent. To test this, we train a second SqueezeNet on CelebA and use (filter) dropout throughout its training ($p_{\text{drop}} = 0.5$). This model is $95.0\%$ accurate on test images, which is slightly lower than the non-dropout model. We compute the Neuron Shapley values in the new model. The values with dropout are more concentrated around zero (Fig. 4(a)). As expected, the dropout Squeezenet is also more robust to the removal of high-value filters (Fig. 4), presumably, because dropout encourages more redundancy. Though it's interesting that even with dropout, removing just the top 30 filters can completely diminish the network's performance, suggesting that there's still a small number of critical neurons.

## B  Proof

The following proof is a direct map of the original Shapley value proof in the cooperative game theory setting [37].

### B.1  Neuron Shapley satisfies the three desired properties

**Zero contribution** If we have a neuron $i$ that contributes nothing to any subset of the rest of neurons, by definition of the Shapley value formula, its value would be zero. **Symmetric Elements** If two neurons contribute exactly the same to any subset of the rest of neurons, again using the Shapley value formulat, they will have the same values by definition. **Additivity** Assume we have $V(.) = V_1(.) + V_2(.)$. It follows that:

$$\phi_i(V) = \frac{1}{|N|} \sum_{S \subseteq N - \{i\}} \frac{V(S \cup \{i\}) - V(S)}{\binom{|N|-1}{|S|}} = \frac{1}{|N|} \sum_{S \subseteq N - \{i\}} \frac{(V_1(S \cup \{i\}) - V_1(S)) + ((V_2(S \cup \{i\}) - V_2(S))}{\binom{|N|-1}{|S|}}$$

$$= \frac{1}{|N|} \sum_{S \subseteq N - \{i\}} \frac{V_1(S \cup \{i\}) - V_1(S)}{\binom{|N|-1}{|S|}} + \frac{1}{|N|} \sum_{S \subseteq N - \{i\}} \frac{V_2(S \cup \{i\}) - V_2(S)}{\binom{|N|-1}{|S|}} = \phi_i(V_1) + \phi_i(V_2) \; \square$$

### B.2  Proof of uniqueness

We show that any contribution scheme $\phi_{i \in N}$ that satisfies the three desired properties is identical to Neuron Shapley.

Consider a simple binary performance metric $w_R$ where for a subset $R$ of the neurons ($R \subseteq N$), we have: $w_R(S \subseteq N) = 1$ if $R \subseteq S$ and $w_R(S) = 0$ otherwise. The contribution scheme $\phi_i(w_R) = \phi_i^R$ has to divide $w(N) = 1$ among players while satisfying the three properties. By definition of the

zero contribution property, a player $i$ wheres $i \notin R$ must have zero contribution. It's also clear that any two players $i \in R$ and $j \in R$ are exchangeable and therefore should have equal contribution. Therefore, the only $\phi_i^w$ that satisfies the conditions is: $\phi_i^w = \frac{1}{|R|}$ if $i \in R$ and $\phi_i^w = 0$ otherwise.

Given all of the subsets $R \subseteq N$ and the simple performance metrics $w_R$ as defined above, we can write any performance metric $V(.)$ as a linear combination of these simple metrics i.e. for any subset $S \subseteq N$:

$$V(S) = \sum_{R \subseteq N} c_R w^R(S)$$

where:

$$c_R = \sum_{T \subseteq R} (-1)^{|R|-|T|} V(T)$$

*Proof.* For an arbitrary $S \subseteq N$ we have:

$$\sum_{R \subseteq N} c_R w^R(S) = \sum_{R \subseteq S} c_R = \sum_{R \subseteq S} \sum_{T \subseteq R} (-1)^{|R|-|T|} V(T) = \sum_{T \subseteq S} \left( \sum_{T \subseteq R \subseteq S} (-1)^{|R|-|T|} \right) V(T)$$

$$= \sum_{T \subseteq S} \left( \sum_{i=|T|}^{|S|} \binom{|S|-|T|}{|S|-i} (-1)^{i-|T|} \right) V(T)$$

the term inside parentheses is the binomial expansion of $(1-1)^{|S|-|T|}$ meaning that it is equal to one if $|T| = |S|$ and zero otherwise. The only case where $|T| = |S|$ while $T \subseteq S$ is when $T = S$. Therefore:

$$\sum_{R \subseteq N} c_R w^R(S) = V(S)$$

$\square$

Now considering that our $\phi_i$ should satisfy additivity in performance metric, for any player $i \in N$ we must have:

$$\phi_i(V) = \sum_{R \subseteq N} c_R \phi_i(w^R)$$

Given the result of the previous lemma, we must have:

$$\phi_i(V) = \sum_{\substack{R \subseteq N \\ i \in R}} \sum_{T \subseteq R} (-1)^{|R|-|T|} \frac{V(T)}{|R|}$$

and by changing the order of summation we have:

$$\phi_i(V) = \sum_{T \subseteq N} \sum_{\substack{R \subseteq N \\ T \cup \{i\} \subseteq R}} (-1)^{|R|-|T|} \frac{\phi_i(T)}{|R|}$$

let's define:

$$\gamma_i(T) = \sum_{\substack{R \subseteq N \\ T \cup \{i\} \subseteq R}} (-1)^{|R|-|T|} \frac{\phi_i(T)}{|R|}$$

for two subsets that only differ in $i$'th filter (i.e. $S_2 = S_1 \cup \{i\}$, we have $\gamma_i(S_2) = -\gamma_i(S_1)$ (as all the right-hand-side terms are the same except for $|T|$). It follows that:

$$\phi_i(V) = \sum_{\substack{S \subseteq N \\ i \in S}} \gamma_i(S)(V(S) - (V(S \setminus \{i\}))$$

and for each $S$ that contains $i$, there are $\binom{|N|-|S|}{|N|-|R|}$ sets of filters such that $S \subseteq R$. We have:

$$\gamma_i(S) = \sum_{i=|S|}^{|N|} (-1)^{i-|S|} \binom{|N|-|S|}{|N|-i} \frac{1}{i} = \sum_{i=|S|}^{|N|} (-1)^{i-|S|} \binom{|N|-|S|}{|N|-i} \int_0^1 x^{i-1} dx$$

$$= \int_0^1 \sum_{i=|S|}^{|N|} (-1)^{i-|S|} \binom{|N|-|S|}{|N|-i} x^{i-1} dx = \int_0^1 x^{|S|-1} \sum_{i=|S|}^{|N|} (-1)^{i-|S|} \binom{|N|-|S|}{|N|-i} x^{i-|S|} dx$$

$$= \int_0^1 x^{|S|-1}(1-x)^{|N|-|S|} dx = \frac{(|S|-1)!(|N|-|S|)!}{|N|!}$$

and finally we have:

$$\phi_i(V) = \sum_{\substack{S \subseteq N \\ i \in S}} \frac{(|S|-1)!(|N|-1S)!}{|N|!}(V(S)-(V(S \setminus \{i\}))$$

$$= \frac{1}{|N|} \sum_{\substack{S \subseteq N \\ i \in S}} \frac{1}{\binom{|N|-1}{|S|-1}}(V(S)-(V(S \setminus \{i\}))$$

$$= \frac{1}{|N|} \sum_{S \subseteq N \setminus \{i\}} \frac{1}{\binom{|N|-1}{|S|}}(V(S \cup \{i\}) - V(S)) \;\square$$

# C   Implementation Details

**Datasets**   For Inception-v3 architecture we use ImageNet dataset. For computing the overall importance of neurons, we use 25000 images of the validation set such that at each iteration of $Alg.$ 1, we sample a batch of 128 random images from this set. We report the results for the remaining 25000 validation images. For class-specific neuron Shapley values, for each class we use 25 validation images and report the results on the remaining 25 validation images of that class. For the Squeezenet architecture and the fairness experiment, we use half of the PPB images to find the neuron Shapley values in Alg. 1. We report the results for the remaining half of images.

**Adversarial attack parameters**   For computing vulnerability neuron Shapley values, we apply projected gradient descent (PGD) adversarail attacks on ImageNet images. We use $_\infty$ attacks with maximum purturbation size of $16/255$. We run the attack for 30 iterations using step size of $0.8/255$.

**Alg 1 Parameters**   We implement Alg. 1 for noth architectures by setting tolerance parameter $\epsilon = 0.0001$ and number of important neurons parameters $k = 100$. For Inception-v3 architecture, we set early truncation parameter $v_T = 20\%$ and for Squeezenet, we use $v_T = 60\%$. For both Inception-V3 and Squeezenet models, , to compute confidence bounds, we use the empirical Bernstein error bounds with parameter $\delta = 0.1$.

**Neuronconductance implementation**   We follow the implementation details in the original paper. For the Inception-v3 architecture, following the original paper, we compute importance scores by averaging neuron conductance scores of 100 randomly selected ImageNet images. For the Squeezenet architecture and the fairness experiment, given that there is no implementation in the original paper, we use half of the PPB dataset images (similar to neuron Shapley experiment). For the vulnerable filters experiment, we use the same adversarial attack algorithm we use for the neuron Shapley experiment. For all experiments, we approximate the integration using Riemann sum of 50 intervals; similar to the original paper and the Integrated Gradients paper [42].

**Computational details**   For more efficient computation, we use parallel processing for our experiments. For both neuron Shapley and neuron conductance experiments, we use a cluster of 100 computing machines each with 12 CPU cores. For Neuron Shapley, we parallelize Alg. 1 by using each machine to do a separate iteration of the algorithm. Given that the iterations of the algorithm are not independent from each other (as they are connected through the updated shapley values and confidence bounds of shapley values), we use a meta-process to update all the machines working in parallel simultaneously about this info. For the neuron conductance experiments, we compute each filter's importance on a separate machine. As a comparison, for the Inception-v3 architecture's overall neuron importance scores, the Neuron Shapley method takes 21 hours to converge while the neuronconductance method takes 19 hours. Comparing the number of passes (forward and backward) between two algorithms, in neuron Shapley, each computation of $V(.)$ on Inception-v3 (i.e. each forward pass) is performed on a batch of 128 random samples (out of 25000 images). By running the algorithm for 3000 iterations, given that most iterations are truncated after removing less than 1500 filters, Neuron-Shapley requires around $4.5 \times 10^7$ forward-passes. For neuron conductance, given the original suggested number of steps of 50 for Riemann approximations, the method requires around $4.6 \times 10^7$ gradients (i.e. forward+backward passes).

# D Early truncation

As mentioned in the main text, one method of approximation is to assign zero marginal contribution to filters that appear early on in a sampled permutation. In Fig.5 we sample random order of players and start removing filters one by one with the sampled order (100 times). As the figures suggest, for any coalition of filters that are not large enough, the performance is completely degraded. For the Inception-v3 model, removing around $10\%$ of its nearly 17000 filters is enough to degrade the model. In our experiments, we approximate the marginal effect of filters by zero whenever the network performance falls below $10\%$. This gives us one order of magnitude speed-up as we will not perform the actual forward pass for more than $90\%$ of the filters in a sampled permutation in Alg. 1. The same happens for the Squeezenet model by removing around one-fifth of the filters (out of nearly 3000 filters in the model).

Figure 5: **Truncation** The figure shows the performance of the two models used throughout this paper as filters are removed randomly (100 removal trajectories). It can be seen that for Inception-v3 mode, removing around $10\%$ of filters will break performance. The same is true for Squeezenet by removing around $20\%$ of filters.

# E Model interpretation through Neuron Shapley

A More complete set of examples of important filters of Inception-V3 model are visuazlied in Fig. 6.

Figure 6: **Inception-V3 important filters**

# F  Squeezenet Interpretation

Similar to Fig. 1, in Fig. 7, we show the most important filter in each layer of the Squeezenet model trained for gender detection task. There are a range of filters that can be interpreted as background color, skin color, face angle, amount of hair, and so forth.

Figure 7: **CelebA important filters**

# G   Alg. 1 Sample Efficiency

To investigate the speed-up effect of multi-armed-bandit trick for computing Shapley values, we run the original monte-carlo Shapley algorithm to compute the importance of filters in the squeezenet model. For both monte-carlo Shapley and TMAB-Shapley algorithms, we use empirical Bernstein [33, 32] error bounds which has the benefit of using the empirical variance of the sampled variable. At iteration $t$ of Alg. 1, for the $i$'th filter that has appeared in $\mathscr{U}$ for $t_i$ iterations, we have:

$$|\phi_i - \text{Empirical AVG}(\phi_i)| \leq \sqrt{\frac{2ln(2/\delta)\text{Empirical VAR}(\phi_i)}{t_i}} + \frac{7R}{3}\frac{ln(2/\delta)}{t_i - 1}$$

with probability at least $1 - \delta$. $R$ is the size of the range of $i$'th filter's marginal contributions. Throughout this work, we assume minimum priors over the filters and therefore fix $R = 1$ for all filters i.e. removing a filter will never result in a more than $100\%$ of drop (or increase) in accuracy. We run both algorithms for an error tolerance of $\epsilon = 0.0001$. Fig. 8 depicts the results: (a) First we show the number of samples each algorithm requires for each filter (for better visualization, we rank filters based on their value). As it is seen, TMAB-Shapley is considerably more sample efficient. On average, it requires $8.7\%$ of the samples required for MC-Shapley; in other words, around 11 times smaller number of forward passes on the model. (b) The histogram of the number of filters versus the number of samples shows that the MAB-Shapley algorithm requires considerably fewer number of samples for most of the filters while requiring a large number of samples for a small group of filters that have values close to that of $k$'th filter. (c) Empirically, it seems like although the TMAB-Shapley is not targeted towards accurate computation of values for all filters, the computed values are very close to the accurate values computed by MC-Shapley method (Rank correlation = 0.988, $R^2 = 0.975$. This shows that empirical Bernstein is returning pessimistic error bounds which could be an interesting direction of research for future work.

Figure 8: **TMAB-Shapley sample efficiency**