[Reviews · NeurIPS 2020]

Review 1

Summary and Contributions: The paper proposes to use Shapley Values to find a small number of hidden neurons that are "most responsible" with respect to a tunable objective, i.e., the overall performance of the network, the recall of the network for one specific class, its bias towards certain data distributions or its sensitivity to adversarial perturbations. The authors show that Shapley Values are more effective at determining the few most relevant neurons compared to Neuron Conductance in a number of experiments in the image domain. Moreover, in order to approximate Shapley Values in a reasonable time on large networks, the paper proposes a novel algorithm that combines Monte-Carlo sampling with a multi-armed bandit.

Strengths: The choice of employing Shapley Values to compute the importance of hidden neurons is well theoretically justified and inherits from related literature in the XAI domain. While the idea of estimating the importance of hidden neurons is not new (see, for example, the cited paper of Neuron Conductance), the idea of using multi-armed bandit for determining the few playes with highest attribution is novel and relevant to the community. Although the method is developed with the idea of determining which are the top-k neurons (rather than estimating the Shapley value for all neurons), I find it interesting that the rank correlation with monte-carlo sampling is so high. I imagine that the same method can also be used to find the few most relevant input features for a given prediction. The paper also shows some interesting phenomena about convolutional neural networks, mainly the fact that there exist a few filters that are responsible for most of the unfair behavior of a classifier or for its sensitivity to adversarial perturbations. Although each phenomenon was not investigated in details, these are still interesting observations in the context of fairness, transparency and security, which are important topics at the moment.

Weaknesses: The idea of applying Shapley values for the understanding of deep neural networks is not new. Several works, such as Lundberg et al., 2017, have already discussed the theoretical motivation for using Shapley values as an attribution method to rank the importance of the input features. Lundberg et al., 2017 also proposed approximations like KernelSHAP and DeepSHAP, which are not compared to TMAB-Shapley. Besides this line of works, the idea of using Shapley values to rank the internal neurons has been proposed by the Stier et al., 2018 (cited) and Florin Leon, 2014 (not cited) in the context of pruning. Finally, Ancona et al., 2019 (not cited) proposed an approximation technique for Shapley values tailored for deep neural networks. How does their method compare with TMAB-Shapley in terms of accuracy and performance? At least a theoretical comparison with Ancona et al. and Lundberg et al. would be needed to justify the authors' claim "this algorithm enables us, for the first time, to compute Neuron Shapley values in large-scale state-of-the-art convolutional networks". I also believe that the experimental section is somewhat superficial. While providing several experiments, the paper does not go in-depth in any of them. This leaves several open questions: - [Neuron Shapley identifies a small number of critical neuron] While it is interesting to see that the network mostly relies on a few very important neurons, what are the practical implications of that? Does it mean that it is possible to use this technique the other way around, and find the least important neurons that can be pruned? This would be a more relevant use case I believe, but it would also require a comparison with other structured pruning techniques. - [Class specific critical neurons] the baselines of comparison are rather weak as one is random and two do not consider how the information is processed after the activation (l2 norm of the weights and l2 norm of the response). I would expect Neural Conductance to be a comparison benchmark for each experiment, in this case, to be shown in Fig 2a. - [Discovering unfair filters] Does the inspection of the "unfair" filters and their activating samples from the training set shed any light on what these filters are capturing? - [Identifying filters vulnerable to adversaries] I did not quite understand the sentence "We note that while the modified network is robust to the original adversary, it is still vulnerable to a new adversary specifically designed to attack the modified network. This requires a white-box adversary who knows exactly which few neurons are zeroed out." Does it mean that the accuracy was tested using the same adversarial images pre-computed for the original network? If it is the case, and if the new network can still be fooled by running PGD on it, how can the method "offer a fast mechanism to repair the model"? The authors would need to show that at least for some attacks (either white or black box) the robustness is improved when the new model is attacked from scratch. The comparison with Neuron Conductance, which is the only strong baseline, seems to be unfair as 25000 images are used to compute Neuron Shapley and only 100 to compute conductance (according to Appendix C). Why this is the case? This means that conductance is computed without providing not even 1 example per class. The reported computational time is also misleading, as conductance can be computed much faster (https://arxiv.org/pdf/1807.09946.pdf, a possible implementation https://captum.ai/docs/algorithms). Since the authors are using a quite unique setup (a large CPU cluster), I would suggest reporting the running times in terms on the number of network evaluations needed (either forward or backward). - [1] Ancona et al., 2019, Explaining Deep Neural Networks with a Polynomial Time Algorithm for Shapley Values Approximation, ICML 2019 - [2] Florin Leon, Optimizing Neural Network Topology Using Shapley Value, 2014, ICSTCC

Correctness: Some claims require clarification: - lines 66-74: 1) what are the "fair credit allocation properties"? According to the original paper, Neuron Conductance also satisfies Zero contribution, Symmetry and Additivity so what are the additional properties that Neuron Shaply satisfies? This section does not mention that gradient-based methods are usually much faster than sampling-based methods. - line 134: while the definition considers all possible coalitions, the marginal contribution of each feature is averaged over all coalitions thus Shapley values is still a linearization of the model and some cross-dependencies might be hidden in the average. For example, while it is guaranteed that a null player will be assigned zero attribution, the other way around is not true: if the Shapley value is 0, the neuron might still have a significant role. Also, the paper says "is one of the few methods that take such interactions into account". What are the other methods? - line 210: there is a vast literature on pruning neurons or filters (structured pruning) like [1-4] - as mentioned before, it is not clear to the reviewer how the results on adversarial examples are computed. [1] HE, Yihui ; ZHANG, Xiangyu ; SUN, Jian: Channel Pruning for Accelerating Very Deep Neural Networks. In: The IEEE International Conference on Computer Vision (ICCV), Oct 2017 [2] HU, Hengyuan ; PENG, Rui ; TAI, Yu-Wing ; TANG, Chi-Keung: Network trimming: A data-driven neuron pruning approach towards efficient deep architectures. In: arXiv preprint arXiv:1607.03250 (2016) [3] LI, Hao ; KADAV, Asim ; DURDANOVIC, Igor ; SAMET, Hanan ; GRAF, Hans P.: Pruning filters for efficient convnets. In: International Conference on Learning Representations (ICLR) (2017) [4] MOLCHANOV, Pavlo ; TYREE, Stephen ; KARRAS, Tero ; AILA, Timo ; KAUTZ, Jan: Pruning convolutional neural networks for resource efficient inference. In: International Conference on Learning Representations (ICLR) (2017)

Clarity: The paper is well written and easy to follow. Some evident typos: - bad-formatting at line 3 of Algorithm 1 - "Postivie" instead of "Positive" in the title of Fig. 1 In general, I would prefer if the comparison with Neural Conductance was inlined in each experimental section, instead of being left in a marginal position.

Relation to Prior Work: Most relevant related works are cited and the differences are discussed, even though some clarification is needed (see above).

Reproducibility: Yes

Additional Feedback: Overall, I find the paper quite interesting, clear and easy to read. The idea of using MAB to find the highest ranking players is nice and I consider this the major contribution of the paper. Shapley values is not a novel idea in the XAI community and it was previously used in the pruning literature for the ranking of hidden neurons, so the novelty is limited in this direction. The experimental section needs better comparison with other methods or at least a fair comparison and discussion of conductance, which should be much faster to compute. ====================================================== After reading the authors' feedback and the other reviews, I increase my score to 7. This is surely an interesting paper. As pointed out by another reviewer, MBA applied to Shapley Values is not entirely new. I still believe the authors should clarify their experiment on adversarial examples as a "black-box attacks" normally refers to a setting where the attacker does not have access to the model parameters but *has* access to the prediction of the model for a given input. In this setting, many "query-based" attacks have been proposed and it does not seem that the experiment in this paper actually runs any of these on the pruned network, thus making the claim potentially misleading.


Review 2

Summary and Contributions: the authors propose using the Shapley value over a game over model elements (neurons/filters or any other type). The value of a subset of neurons can be a lot of things, like model accuracy, model fairness or anything else really. The authors suggest a multi-armed bandit approach to compute the Shapley value (this is useful in applications where one just wants to identify the more important elements rather than knowing their precise Shapley value) and evaluate their approach on a variety of datasets. The results show that removing the neurons identified as important by the approach indeed results in a significant drop in network accuracy. Similarly, removing neurons responsible for bad model behavior results in significant improvement. This is useful since the number of neurons that need to be removed is typically small.

Strengths: This is an interesting application of the Shapley value to ML, which makes sense in the domain. The usage of a multi-armed bandit approach speeds up the computation significantly and this trick might even be useful in other situations where Shapley values are calculated The experimental results, including the potential application to fairness seem promising. The most interesting thing about the paper is the authors experimentally show how successful in different tasks, especially finding out class-specific crucial filters. It’s surprising when the proposed method can find filters that affect only one target class with little changing to the overall accuracy. This will help understand the network better.

Weaknesses: Some of the experimental results need more details and elaboration. Top-k crucial filters: the authors did not yet explain the interaction between these filters as mentioned in line 65. How are these filters related to each other? Are they from different layers or the same layer? Robustness experiment: In line 275, when the authors mention ‘original adversary’, do they mean the PGD attack (i.e. rerunning the attack on the modified network) or just the adversarial images generated beforehand? If the latter, the result is much less surprising. What is strange about the paper is the evaluation method. First, the authors need to estimate Shapley values for individual filters from performance of coalitions. Then for most experiments, they measure the performance of the method by zeroing out the set of top k filters. If we are interested in top k crucial filters, is there any way to find them other than via computing Shapley value? Feature selection methods seem like a good candidate for comparison, not just objective performance. I would like to point out that https://arena.gov.au/assets/2019/06/consort-reward-structures.pdf discusses a similar approach for Shapley value computations using MAB.

Correctness: The claims about the achieved speedup are a bit vague and only evaluated on two datasets. Which seems more like a proof of concept than a thorough investigation.

Clarity: Yes overall, with a few typos and minor issues: see additional comments. There are minor issues with typos and Algorithm 1 is ill-defined. The authors use the terms “neuron” and “filter” interchangeably, which is confusing, and calling the methods “Neuron shapley” feels misleading.

Relation to Prior Work: yes

Reproducibility: Yes

Additional Feedback: line 21-23: sentence fragment, Line 26: neruons -> neurons Line 134 “This, to our knowledge, is one of the few methods that take such interactions into account.” → There are actually several such methods in the Game Theory literature. For example the Banzhaf interaction index (see Hammer and Holtzmann “Approximations of Pseudo-Boolean Functions”) line 137: v: 2^n \to R Line 142 appendix b → Appendix B (similar in other lines) Line 155 literature. [29, 6, 28] → literature [29, 6, 28]. Line 180 our three approximation methods, → counting Monte Carlo as “our” is a bit of a stretch Line 200 This statement is very unclear, given that TMAB already includes the truncation. What does the 11.5 in Line 194 mean is this only from the Bandit approach? Algorithm 1 Line 3: Something went wrong after sigma Algorithm 1 is unclear and/or wrong v is not clearly defined and or initialized line 10-11 make no sense without initialisation Line 274-277 This statement is very vague. Also robustness against a specific attack is practically useless. See “On Evaluating Adversarial Robustness” Carlini et al. 2019, for guidelines on testing adversarial robustness


Review 3

Summary and Contributions: This paper introduces a bandits-based algorithm to quantify the importance of each filter and/or neuron in a network. The authors bridge the idea of the Shapley value from cooperative game theory in order to devise a saliency measure that takes into account the interdependencies/interactions between the neurons, and more generally, components of the network. The authors then use and build upon ideas from bandits literature in their algorithm to tame the intractability of exact Shapley value computations, and to identify the neurons with the highest Shapley values (i.e., most important neurons).

Strengths: - The paper is of high relevance as it attempts to remedy a notable shortcoming of contemporary work on pruning -- capturing the complicated interactions between the weights/neurons/filters of a network when assigning importances to or understanding the role of each neuron/filter - The idea of using a bandits-based algorithm for pruning is novel and very interesting - Nice applications and empirical evaluations to fairness and adversarial attacks, and some compelling results

Weaknesses: - Lack of theoretical guarantees: although the bandits-based algorithm is interesting and a principled approach to this problem, I was not convinced that it was better than MC sampling (with an importance sampling distribution) since it has theoretical bounds on the accuracy of the approximations. The authors state that bandits-based should fare better since most of the Shapley Values will be close to 0 (i.e., most neurons are not that important), but there is no regret bound given for the bandits-based algorithm and Fig. 3 (a) doesn’t prove that the *total* number of samples required by the algorithm is smaller. - Lack of clarity and detail in the method and background: I would have liked to see more of an explicit connection to bandits literature (e.g., the rewards in our case are so and so, the arms correspond to different neurons, the rewards are… and are bounded in [0,1] because...). It is also not clear what the computational cost of the algorithm is and how it compares to MC sampling. Since the algorithm has a while loop, it is not clear whether the terminating condition is always reached or what the (expected) number of iterations is. It is also not clear how the algorithm scales with the input k (the number of neurons to “keep”). Can it be so large as to be used in network pruning so that only the essential filters are kept? Update: I have raised my score to a 6 in light of the reviewer discussion and the authors' rebuttal. I still think that the work can be improved by using existing work in MAB for pure exploration (e.g., see [1]), and I think that trying to describe a properietary algorithm rather than refer to well-established MAB work hurts the exposition and makes it difficult to understand. Given the discussion, I am more convinced that the tools used in this work -- MAB & Shapley values --, although not novel, may be of interest to the pruning community in particular, since the main challenge in pruning is in dealing with parameter interdepencies in a computationally efficient way. I would encourage the authors to emphasize this connection and potential application to pruning in their work. [1] https://tor-lattimore.com/downloads/book/book.pdf , Chapter 33 (pg. 409)

Correctness: My biggest qualm with the claims in this paper is the lack of quantitative results and comparisons to other neuron importance measures, especially those that were developed for use in network pruning (e.g.,[1-3]). The author did note that a future direction is to use their algorithm in a prune-retrain (or change-retrain) pipeline, which is fine, but even a simple prune-only experiment that demonstrates the value of the bandits-based approach with Shapley values in retaining only the important filters of a network would make the paper much more impactful

Clarity: The paper is well-written for the most part, but I would have appreciated more details and clarity in the description of the method and the pseudocode.

Relation to Prior Work: For the most part, there is a nice exposition of related work and the authors justify each component of their algorithm by relating it to prior work. However, some claims in relation to previous work, such as the superiority of bandits-based vs. MC sampling and the empirical or theoretical accuracy of the Shapley Value in contrast to previously introduced neuron importance metrics could have used more justification (either theoretically or experimentally)

Reproducibility: Yes

Additional Feedback:

[Author Response · NeurIPS 2020]

We would like to thank all the reviewers for their insightful comments and for finding this work interesting overall. We'll carefully incorporate your helpful suggestions into the paper revision.

***Reviewer#1*** Thank you for your very thoughtful feedback. We benefit a lot from it. We agree that there has been several interesting works applying Shapley values for different aspects of interpretability. We tried to discuss the most relevant literature and will add more discussion of the missing citations. One distinction we'd like to draw is that our method is designed for interpreting the individual elements/filter of the model while most the existing literature is designed for computing feature-importance scores. As discussed in the paper, we are aware that such methods (including Ancona et al and Lundberg et al) can be extended for computing neuron-importance, even though it was not used this way in the original papers. Assume each layer to be the input of the model and find the importance of neurons in that layer using one of these methods. Then repeat this process for every layer and get importance scores for all neurons. However, one of our desiderata is that the scores should incorporate the joint interactions of elements in different layers. Existing methods will only look at joint behavior of elements that are in the same layer. Directly modeling the network as a cooperative game between all of the neurons is a principled framework to account for all of the neuron interactions. Similar idea was also explored by Stier et al and Florin Leon for pruning small neural networks and our MAB approach enables us to compute scores for very large DNNs. **As the reviewer requested, we have added comparisons with Neuron Conductance for Fig2 during the response.** Neuron Shapley has a superior performance in finding the class-specific neurons. For conductance, removing top-10 and top-20 class specific neurons for the "Carousel" class reduced accuracy to $64\%$ and $40\%$ compared to $20\%$ and $8\%$ using Shapley (both methods use 25 carousel images). We will revise Fig 2 to add these and other comparisons. **Regarding model repair for adversarial attacks**, our results strengthen the overall message of the paper – adversarial vulnerability can be attributed to a small subset of neurons. Removing this subset substantially improved black-box robustness. While white-box attack is harder to defend, in many applications adversaries only have black-box access and pruning can still be an effective repair. Overall our results highlight how the sparsity of Neuron Shapley values can facilitate model repair (which has not been explored as much) and interpretation. Model reduction through pruning is a different application is an interesting future direction of research (and there's some work on this) which is separate from the scope of our current work. The reviewer suggested **comparing the number of passes in the model between neuron Shapley and conductance**. We will add this to the revision. Overall the two methods require the similar number of passes. In neuron Shapley, each computation of $V(.)$ on Inception-v3 (i.e. each forward pass) is performed on a batch of 128 random samples (out of 25000 images). By running the algorithm for 3000 iterations, given that most iterations are truncated after removing less than 1500 filters, Neuron-Shapley requires around $4.5 \times 10^7$ forward-passes. For neuron conductance, given the original suggested number of steps of 50 for Riemann approximations, the method requires around $4.6 \times 10^7$ gradients (i.e. forward+backward passes). The Shrikumar et al's paper mentioned by the reviewer has similar computational complexity to our implementation of neuron-conductance work. The reviewer mentions the discrepancy in the number of images used for neuron-conductance and Shapley. We used 100 images for conductance because that's what was used in the original work. Moreover, the number of images used for Shapley led to the two methods having the same computational cost. **For the response, we ran neuron-conductance with** $200$ **images and found no significant improvement.** For Inception-v3 model, removing 57 filters reduced accuracy to random (from 59 using 100 images). For the fairness experiment, removing the top 105 unfair filters, accuracy on PPB increased to $88.5\%$ ($88.7\%$ using 100 images). The Shapley results are still substantially better.

***Reviewer#3*** Thanks for your insightful review and support of the paper! Regarding the location and interaction of the important filters, Fig.1 (lower panel) shows how many important neurons are in each layer. We'll extend these results and include statistics of important neurons for other experiments. It's an interesting question for other ways to find the top-$k$ filters. There are not many principled approaches for doing this to the best of our knowledge. There are strategies using gradient methods to assign neuron importance; conductance is a SOTA method in this class. Our experiments demonstrate that Shapley substantially outperforms conductance. Moreover Shapley uniquely satisfies several desirable mathematical properties. Thanks for mentioning the "Reward Structures" work; we'll cite and discuss this work.

***Reviewer#4*** Thank you for your positive comments and helpful suggestions. It's a great advice to make better connections to the bandit literature. We'll incorporate this in the revision and will edit the description of algorithm in our work to make this connection clear. We can also add theoretical guarantees for our algorithm. TMAB-Shapley is solving a top-k arm selection problem where each arm's distribution is bounded ( between -1 and1) for which; we can directly adapt the regret bounds from the MAB literature to analyze our method. The reviewer mentions comparison with alternative neuron importance measures. We have provided comparisons with the Neuron-Conductance method which is the best performing of the existing neuron-importance measures. Shapley performed substantially better than conductance. We have also added more comparisons to conductance in the response (please see the last few comments to Reviewer 1). Your suggestion to combine top-k arm selection and static model-pruning is very interesting and is great for further research. This first work is mostly focused on applications of Neuron Shapley for model-repair and interpretability (rather than pruning) and our experiments are designed for these use-cases.

[Meta-Review · NeurIPS 2020]

This paper develops an efficient multi-arm bandit method for determining the most useful nodes in a network. Interesting applications for improving fairness and possibly increasing robustness to adversarial attacks are mentioned. Reviewers provided three insightful reviews mentioning the pros (a cool combination of techniques with encouraging results worth the field hearing about) and cons (not the most novel method and more comparison with other methods would be beneficial) of paper acceptance. There is weak but unanimous support for acceptance.